# Protective Effects of BPC 157 on Liver, Kidney, and Lung Distant Organ Damage in Rats with Experimental Lower-Extremity Ischemia–Reperfusion Injury

**DOI:** 10.3390/medicina61020291

**Published:** 2025-02-08

**Authors:** Hüseyin Demirtaş, Abdullah Özer, Alperen Kutay Yıldırım, Ali Doğan Dursun, Şaban Cem Sezen, Mustafa Arslan

**Affiliations:** 1Department of Cardiovascular Surgery, Faculty of Medicine, Gazi University, 06560 Ankara, Turkey; drhuseyindemirtas@yahoo.com (H.D.); dr-abdozer@hotmail.com (A.Ö.); drayildirim@yahoo.com (A.K.Y.); 2Department of Physiology, Faculty of Medicine, Atılım University, 06830 Ankara, Turkey; alidogandursun@gmail.com; 3Vocational School of Health Services, Atılım University, 06805 Ankara, Turkey; 4Home Care Services, Medicana International Ankara Hospital, 06520 Ankara, Turkey; 5Department of Histology and Embryology, Faculty of Medicine, Kırıkkale University, 71450 Kırıkkale, Turkey; sezenscem@gmail.com; 6Department of Anesthesiology and Reanimation, Faculty of Medicine, Gazi University, 06560 Ankara, Turkey; 7Application and Research Centre for Life Sciences, Gazi University, 06560 Ankara, Turkey; 8Centre for Laboratory Animal Breeding and Experimental Research (GÜDAM), Gazi University, 06560 Ankara, Turkey

**Keywords:** BPC 157, ischemia–reperfusion, lower extremity, lung, kidney, renal, TAS, TOS

## Abstract

*Background and Objectives*: Ischemia–reperfusion (I/R) injury can affect multiple distant organs following I/R in the lower extremities. BPC-157’s anti-inflammatory and free radical-neutralizing properties suggest its potential in mitigating ischemia–reperfusion damage. This study evaluates the protective effects of BPC-157 on remote organ damage, including the kidneys, liver, and lungs, in a rat model of skeletal muscle I/R injury. *Materials and Methods*: A total of 24 male Wistar albino rats were randomly divided into four groups: sham (S), BPC-157(B), lower extremity I/R(IR) and lower extremity I/R+BPC-157(I/RB). Some 45 min of ischemia of lower extremity was followed by 2 h of reperfusion of limbs. BPC-157 was applied to groups B and I/RB at the beginning of the procedure. After 2 h of reperfusion, liver, kidney and lung tissues were harvested for biochemical and histopathological analyses. *Results*: In the histopathological examination, vascular and glomerular vacuolization, tubular dilation, hyaline casts, and tubular cell shedding in renal tissue were significantly lower in the I/RB group compared to other groups. Lung tissue showed reduced interstitial edema, alveolar congestion, and total damage scores in the I/RB group. Similarly, in liver tissue, sinusoidal dilation, necrotic cells, and mononuclear cell infiltration were significantly lower in the I/RB group. Additionally, the evaluation of TAS, TOS, OSI, and PON-1 revealed a statistically significant increase in antioxidant activity in the liver, lung, and kidney tissues of the I/RB group. *Conclusions*: The findings of this study demonstrate that BPC-157 exerts a significant protective effect against distant organ damage in the liver, kidneys, and lungs following lower extremity ischemia–reperfusion injury in rats.

## 1. Introduction

Ischemia–reperfusion (IR) injury is a complex pathological process that occurs when blood supply to an organ is temporarily interrupted and subsequently restored, resulting in significant tissue damage and dysfunction [1]. During the ischemic phase, the deprivation of oxygen and essential nutrients leads to profound cellular dysfunction, metabolic disturbances, and potential structural damage [2]. Paradoxically, the restoration of blood flow during reperfusion exacerbates the injury through mechanisms involving oxidative stress, inflammation, and endothelial dysfunction [3]. This dual-phase injury not only impacts the initially ischemic tissue but also induces systemic responses that contribute to distant organ damage, particularly in the liver, kidneys, and lungs [4]. The systemic release of reactive oxygen species (ROS), proinflammatory cytokines, and other mediators amplifies multi-organ injury, leading to increased morbidity and mortality in various clinical contexts, including surgery, trauma, and vascular diseases [5,6]. A comprehensive understanding of IR injury mechanisms is vital for developing effective therapeutic strategies to mitigate both local and systemic tissue damage.

During ischemia, oxygen deprivation forces cells to rely on anaerobic metabolism, resulting in ATP depletion and impaired function of ion pumps such as Na⁺/K⁺-ATPase, leading to ionic imbalances [7]. The consequent intracellular calcium overload, driven by the Na/Ca exchanger, further disrupts cellular homeostasis [1]. Upon reperfusion, the sudden influx of oxygen triggers excessive production of ROS—such as superoxide anions, hydroxyl radicals, and peroxynitrite—which inflict damage on microvascular structures and mitochondrial membranes [8]. This mitochondrial injury promotes the release of apoptotic mediators like cytochrome c and caspases, initiating programmed cell death. Additionally, enzymes such as xanthine oxidase and nitric oxide synthase are activated, generating ROS that react with nitric oxide (NO) to form peroxynitrite, further aggravating oxidative stress. Hypoxia-induced inhibition of propylhydroxylase enzymes leads to post-translational modifications in nucleic acids and proteins [1]. The accumulation of ROS overwhelms endogenous antioxidant defenses, creating a vicious cycle of oxidative damage. Neutrophil activation and adhesion to endothelial cells amplify the inflammatory response, resulting in endothelial dysfunction and exacerbating tissue injury [8].

The systemic consequences of IR injury extend beyond the primary ischemic site, affecting distant organs such as the lungs, kidneys, and liver [4,8,9,10]. Lung tissue is highly vulnerable to injury following limb IRI. Reperfusion can lead to acute lung injury (ALI) or acute respiratory distress syndrome (ARDS), characterized by increased vascular permeability, pulmonary edema, and inflammation [11,12]. The kidneys are similarly vulnerable, often developing acute kidney injury (AKI), which can further deteriorate the patient’s condition [4]. Ischemic postconditioning has been explored as a strategy to alleviate limb IR-induced renal injury, with studies revealing notable histopathological alterations in kidney tissues post injury [13]. The liver, integral to detoxification and metabolism, is also susceptible to oxidative and inflammatory damage during systemic IR injury, impairing its vital functions [4]. Although direct evidence linking limb IR to hepatic injury is limited, the interconnected nature of organ systems implies that hepatic dysfunction can occur secondary to systemic inflammatory responses. The involvement of these distant organs in IR injury significantly complicates clinical outcomes, increasing the risk of multi-organ failure and underscoring the need for targeted therapeutic interventions.

BPC 157, a stable pentadecapeptide derived from human gastric juice and a key component of the body protection compound, has garnered attention for its therapeutic potential [14]. Clinical trials in patients with inflammatory bowel disease (IBD) have demonstrated its safety profile, with no lethal dose (LD_1_) identified [15,16]. Although its precise mechanisms remain partially understood, BPC 157 has shown substantial protective effects across various tissues, including the skin [17,18], gastrointestinal tract [19,20,21,22], tendons [23,24], ligaments [25], muscles [26,27], bones [28], myocardium [29], and blood vessels [30,31]. Notably, BPC 157 modulates several biological pathways, particularly the nitric oxide (NO) system [19,31,32,33]. It mitigates oxidative damage by scavenging free radicals and upregulating antioxidant enzymes, thereby preserving cellular homeostasis and preventing apoptosis [22,34,35,36,37]. Its antioxidant properties have proven effective in reducing the oxidative stress and tissue damage associated with IR injury [38,39,40].

Experimental studies have demonstrated reperfusion injury in distant organs following lower limb ischemia–reperfusion [9,41,42,43]. However, the majority of research has predominantly focused on localized organ damage, with limited exploration of distant organ involvement. Considering the widespread systemic effects of IR injury, identifying protective interventions is imperative [3]. Animal studies have highlighted BPC 157’s protective effects across various organ systems [21,22,23,25,27,29,30,34,36,38,39,44]. Despite these promising findings, the protective role of BPC 157 in distant organs following limb IR injury remains insufficiently explored. This study aims to investigate the protective effects of BPC 157 on remote organ damage, specifically the kidneys, liver, and lungs, in a rat model of skeletal muscle I/R injury.

## 2. Material and Method

### 2.1. Animals

Study protocols were conducted in male Albino Wistar rats that had a weight of 250–350 g, were 12 weeks old, and were bred in-house at the animal facility of Gazi University Experimental Research Center, Ankara, Turkey. All experimental procedures were carried out at the Gazi University Animal Laboratory following ethical guidelines approved by the Gazi University Experimental Animal Ethics Committee (G.Ü.ET-24.075). The rats were individually housed in a controlled environment, with the temperature maintained at 20–21 °C and humidity levels at 45–55%. A 12 h light/dark cycle was provided, and the animals had unrestricted access to food and distilled water.

### 2.2. Drug

The administration of drug followed protocols from previous studies [22,32,33,38,45], without incorporating a carrier or peptidase inhibitor, for the stable gastric pentadecapeptide BPC 157. This peptide, a fragment of the human gastric juice protein BPC, is highly soluble in water at pH 7.0 and in saline. BPC 157 (sequence: GEPPPGKPADDAGLV; molecular weight: 1419.54 g/mol; and obtained from Sigma-Aldrich, Taufkirchen, Germany) was synthesized with a purity exceeding 95%, as confirmed through high-performance liquid chromatography (HPLC). Both the dosage and method of administration were determined based on earlier studies, with the application dose of BPC 157 set at 20 μg/kg, as indicated in the References [22,32,33,38,45].

### 2.3. Experimental Protocol

A total of 24 rats were randomly divided into four experimental groups (n = 6 per group): control, group B, IR, and IR-BPC157. All surgical procedures were conducted under general anesthesia, induced via intramuscular injection of 50 mg/kg ketamine hydrochloride (500 mg/10 mL; Ketax^®^; Vem İlaç San. Tic. A.Ş. İstanbul, Turkey) and 10 mg/kg xylazine hydrochloride (Alfazyne^®^ vial 2%; Ege Vet, Ltd., İzmir, Turkey). If any signs of surgical stress or response to tail pinch were observed, additional doses of 20 mg/kg ketamine and 5 mg/kg xylazine were administered to maintain anesthesia. After 30 min, the procedure began with the animals positioned supine under a warming lamp.

In the control group, saline (0.3 cc, 0.9%) was administered intraperitoneally 45 min post-midline laparotomy, which was the sole surgical intervention. After a 2 h observation period, the animals were euthanized under anesthesia, and lung, kidney, and liver tissues were collected. In group B, BPC 157 (20 μg/kg) was injected intraperitoneally 45 min after the laparotomy, with no further interventions. The rats were euthanized after a 2 h follow-up, and lung, kidney, and liver samples were taken. In the IR group, an atraumatic microclamp was applied to the infrarenal aorta for 45 min, followed by a 120 min reperfusion period after clamp removal. The animals were subsequently sacrificed, and tissue samples from the lungs, kidneys, and liver were harvested. In the IR-BPC group, BPC 157 (20 μg/kg) was administered intraperitoneally after the laparotomy, followed by clamping of the infrarenal aorta for 45 min. Once the clamp was removed, reperfusion was permitted for another 120 min, after which the animals were euthanized, and the lung, kidney, and liver tissues were collected.

All rats were euthanized by collecting 5–10 mL of blood from the abdominal aorta following the administration of ketamine (100 mg/kg) and xylazine (10 mg/kg). Death was confirmed by observing the cessation of heartbeat and respiration, after which the animals were monitored for an additional 2 min to ensure complete demise.

### 2.4. Histopathological Evaluation

Histopathological assessments were performed in the Department of Histology at Kirikkale University, Kirikkale, Turkey. The tissues were fixed in 10% formalin for 48 h at room temperature. After fixation, the specimens were embedded in paraffin and subjected to routine tissue processing. Tissue sections that were 4 μm thick were sliced from the paraffin blocks using a microtome (Leica RM2245, Leica Microsystems, Wetzlar, Germany) and stained with hematoxylin and eosin for histopathological evaluations.

For examination of prepared specimens, tissue sections of each group were prepared. The sections were scanned from end to end, and 1–2 images were taken from appropriate areas. Each field of view was scanned once.

The tissue sections were examined under a light microscope (Leica DM 4000 B, Leica Microsystems, Wetzlar, Germany) connected to a computer. Photographs of the samples were taken using Leica LAS V4.9. Hematoxylin and eosin-stained tissue sections were examined at 200× and 400× magnifications.

#### 2.4.1. Liver

Each sample was examined for hydropic degeneration, sinusoidal dilatation, pycnotic nuclei, necrosis, and mononuclear cellular infiltration in the parenchyma. The semiquantitative evaluation technique used by Abdel-Wahhab et al. [46] for histological testing was applied to interpret the structural changes in the hepatic tissues of the control and treatment groups, with a negative point (0) representing no structural changes, one positive point (1, +) indicating mild changes, two positive points (2, ++) representing medium structural changes, and three positive points (3, +++) indicating severe structural changes [47].

#### 2.4.2. Renal

Renal injury was evaluated by assessing glomerular vacuolization (GV), tubular dilation (TD), vascular vacuolization and hypertrophy (VVH), tubular cell degeneration and necrosis (TCDN), Bowman space dilation (BSD), tubular hyaline cylinders (THC), lymphocyte infiltration (LI), and tubular cell shedding (TCS) [48]. Renal injuries were scored as follows: 0, no change; +1, minimal change; +2, medium; and +3, severe [49].

#### 2.4.3. Lung

In each specimen, neutrophil infiltration and alveolar thickness were assessed to determine the severity of the lung injury. The severity of the injury indicated by each parameter was categorized as falling into one of four levels: none, minimal (+), moderate (++), or severe (+++). The individual scores for each parameter were totaled and recorded as the total injury score [50].

### 2.5. Biochemical Evaluation

The lung, kidney, and the liver samples were quickly frozen in liquid nitrogen and then kept at −80 °C in a deep freezer until they were analyzed for total antioxidant status (TAS), total oxidant status (TOS), oxidative stress index (OSI), and paraoxonase-1 (PON-1).

The tissue analysis procedures were conducted swiftly to prevent thawing. Initially, 22 lancets were employed to extract tissue portions (80–100 mg), which were subsequently weighed and crushed in liquid nitrogen. The resulting powder was transferred to a homogenization tube, where it was mixed with a potassium chloride solution (140 mM) in a 1:10 weight/volume dilution. To maintain a low temperature during homogenization, the tube was placed in a beaker of ice and processed with a homogenizer at 50 rpm for two minutes. The homogenate was centrifuged at 3000× *g* for 10 min, and the supernatants were collected.

Total antioxidant status (TAS) was assessed using a commercial TAS assay kit (RelAssay Diagnostics, Gaziantep, Turkey). A 30 μL sample was combined with 500 μL of measurement buffer, and the absorbance was measured at 660 nm. Following the addition of a colored reagent and a 5 min incubation at 37 °C, a second absorbance reading was taken. TAS levels were calculated using standard Trolox equivalents. Total oxidative status (TOS) was measured using a commercial TOS assay kit (RelAssay Diagnostics, Gaziantep, Turkey). A 75 μL sample was mixed with a buffer, and absorbance was first measured at 530 nm. After the addition of a pro-chromogenic solution and a 5 min incubation at 37 °C, a second absorbance was recorded. TOS levels were determined using standard hydrogen peroxide equivalents. All measurements were conducted in triplicate, and mean values were reported. The levels were examined following the methodology outlined in prior studies [51].

The oxidative stress index (OSI) was calculated using the following formula:

OSI (arbitrary units, AU) = (TOS (μmol H_2_O_2_ equivalent/L))/(TAS (μmol H_2_O_2_ equivalent/L)) × 100

The spectrophotometric measurement of paraoxonase (PON-1) activity was performed using a commercial PON-1 assay kit (RelAssay Diagnostics, Gaziantep, Turkey). PON-1 values were assessed utilizing the approach described in previous research publications [52,53]. The rate of paraoxon hydrolysis (diethylpnitrophenylphosphate in 50 mM glycine/NaOH, pH 10.5 containing 1 mM CaCl_2_) was determined by measuring the increase in absorption at 412 nm at 37 °C. The amount of generated p-nitrophenol was determined from the molar absorption coefficient at pH 8.5, which was 18.290 M^−1^ cm^−1^ at pH 10.5. The quantity of enzyme required to catalyze the hydrolysis of one μmol of substrate at 37 °C was defined as one enzyme unit (U/L).

### 2.6. Statistical Analysis

Statistical analysis was performed using SPSS (version 26.0; IBM Corp., Armonk, NY, USA). The distribution of the data was assessed using the Shapiro–Wilk test. Depending on the data characteristics, Dunn’s test or a one-way ANOVA (for biochemical parameters) was applied, followed by Bonferroni post-hoc analysis. A *p*-value of less than 0.05 was considered statistically significant. Results are presented as mean ± standard error, contingent on the overall distribution of the variables.

## 3. Results

### 3.1. Biochemical Results

#### 3.1.1. Renal Tissue

When the groups were compared in terms of TAS levels in renal tissue, a significant difference was observed between the groups (*p* = 0.019). The TAS level was found to be significantly lower in the IR group compared to the control and BPC-applied groups (*p* = 0.002, *p* = 0.034, respectively). Additionally, the TAS level in the IR-BPC group was significantly higher than that of the control group (*p* = 0.047), (Table 1).

A significant difference was detected between the groups in terms of TOS levels in renal tissue (*p* = 0.014). The TOS levels in the IR group were notably higher compared to those in the control and BPC groups (*p* = 0.002, *p* = 0.022, respectively). Furthermore, the TOS levels in the IR-BPC group were significantly lower when compared to the IR group (*p* = 0.045), (Table 1).

When the groups were compared in terms of OSI levels in kidney tissue, a significant difference was observed between the groups (*p* < 0.001). The OSI level in the IR group was found to be significantly higher compared to the control and BPC groups (*p* < 0.001, *p* = 0.001, respectively). Similarly, the OSI level in the IR-BPC group was significantly higher than that of the control group (*p* = 0.024). Additionally, the OSI level in the IR-BPC group was significantly lower compared to the IR group (*p* = 0.016), (Table 1).

In terms of PON1 enzyme activity in kidney tissue, a significant difference was also detected between the groups (*p* < 0.001). PON1 enzyme activity was significantly lower in the IR group compared to the control and BPC groups (*p* < 0.001, *p* = 0.001, respectively). Likewise, PON1 enzyme activity in the IR-BPC group was significantly lower compared to the control group (*p* < 0.001). However, PON1 enzyme activity in the IR-BPC group was significantly higher than in the IR group (*p* = 0.040), (Table 1).

#### 3.1.2. Lung Tissue

When the groups were compared in terms of TAS (total antioxidant status) levels in lung tissue, a significant difference was observed between the groups (*p* = 0.017). TAS levels in the IR group were significantly lower compared to the control and BPC groups (*p* = 0.002, *p* = 0.042, respectively). Moreover, the TAS level in the IR-BPC group was found to be significantly higher than in the IR group (*p* = 0.034), (Table 2).

A significant difference was also found between the groups in terms of TOS (levels in lung tissue) (*p* < 0.001). The TOS levels in the IR group were significantly higher compared to the control and BPC groups (*p* < 0.001, *p* = 0.001, respectively). Additionally, the TOS levels in the IR-BPC group were significantly lower than those in the IR group (*p* < 0.001), (Table 2).

In terms of OSI levels in lung tissue, a significant difference was observed between the groups (*p* < 0.001). OSI levels in the IR and IR-BPC groups were significantly higher compared to the control and BPC groups (*p* < 0.001, *p* = 0.001, respectively). However, OSI levels in the IR-BPC group were significantly lower than those in the IR group (*p* < 0.001), (Table 2).

When comparing PON1 enzyme activity in lung tissue, a significant difference was detected between the groups (*p* = 0.013). PON1 enzyme activity was found to be significantly lower in the IR group compared to the control and BPC groups (*p* = 0.002, *p* = 0.012, respectively). However, PON1 enzyme activity in the IR-BPC group was significantly higher than in the IR group (*p* = 0.016), (Table 2).

#### 3.1.3. Liver Tissue

When the groups were compared in terms of TAS levels in liver tissue, a significant difference was observed among the groups (*p* = 0.002). TAS levels in the IR and IR-BPC groups were found to be significantly lower compared to the control and BPC groups (Table 3).

A significant difference was also detected between the groups regarding TOS levels in liver tissue (*p* < 0.001). The TOS level in the IR group was significantly higher compared to the control and BPC groups (*p* < 0.001, *p* = 0.001, respectively). Additionally, the TOS level in the IR-BPC group was significantly lower compared to the IR group (*p* < 0.001), (Table 3).

In terms of OSI levels in liver tissue, there was a significant difference between the groups (*p* < 0.001). The OSI level in the IR group was significantly higher compared to the control and BPC groups (*p* < 0.001, all). Similarly, the OSI level in the IR-BPC group was significantly higher than that in the control group (*p* = 0.017). However, OSI levels in the IR-BPC group were significantly lower compared to the IR group (*p* < 0.001), (Table 3).

Regarding PON1 enzyme activity in liver tissue, a significant difference was observed among the groups (*p* < 0.001). PON1 enzyme activity in the IR group was significantly lower compared to the control and BPC groups (*p* < 0.001, all). Similarly, PON1 enzyme activity in the IR-BPC group was significantly lower than in the control and BPC groups (*p* = 0.014, *p* = 0.026, respectively). However, PON1 enzyme activity in the IR-BPC group was significantly higher compared to the IR group (*p* = 0.018), (Table 3).

### 3.2. Histopathological Results

#### 3.2.1. Renal Tissue

Upon light microscopy analysis, significant differences were observed among the groups in various histopathological parameters. Glomerular vacuolization (GV) was found to differ significantly between the groups (*p* = 0.012). GV was significantly more pronounced in the IR group compared to the control and BPC groups (*p* = 0.004 for both). However, GV was significantly lower in the IR-BPC group compared to the IR group (*p* = 0.015) (Table 4, Figure 1).

Tubular dilatation (TD) also showed significant differences between the groups (*p* = 0.023). TD was more frequent in the IR group compared to the control and BPC groups (*p* = 0.007 for both). In contrast, TD was significantly reduced in the IR-BPC group compared to the IR group (*p* = 0.046) (Table 4, Figure 1).

Vascular vacuolization and hypertrophy (VVH) were significantly different across the groups (*p* = 0.040). VVH was observed more frequently in the IR group compared to the control and BPC groups (*p* = 0.013 for both). In the IR-BPC group, VVH was significantly lower than in the IR group (*p* = 0.042) (Table 4, Figure 1).

Tubular cell degeneration and necrosis (THDN) showed no significant differences among the groups (*p* = 0.121) (Table 4, Figure 1).

Bowman space dilatation (BSD) revealed significant differences between the groups (*p* = 0.003), with BSD being more prominent in the IR group compared to the control and BPC groups (*p* = 0.001 for both). In the IR-BPC group, BSD was significantly reduced compared to the IR group (*p* = 0.004) (Table 4, Figure 1).

Tubular hyaline casts (THCs) differed significantly among the groups (*p* = 0.019). THCs were more frequently observed in the IR group compared to the control and BPC groups (*p* = 0.004 and *p* = 0.012, respectively). In the IR-BPC group, THCs were significantly lower than in the IR group (*p* = 0.040) (Table 4, Figure 1).

Lymphocyte infiltration (LI) did not show significant differences between the groups (*p* = 0.071) (Table 4, Figure 1).

Tubular cell shedding (TCS) was significantly different among the groups (*p* = 0.027). TCS was more prominent in the IR group compared to the control and BPC groups (*p* = 0.009 for both). However, TCS was significantly reduced in the IR-BPC group compared to the IR group (*p* = 0.032) (Table 4, Figure 1).

#### 3.2.2. Lung Tissue

Significant differences were observed among the groups regarding interstitial edema in lung tissue (*p* = 0.027). Interstitial edema was more prominent in the IR group compared to the control and BPC groups (*p* = 0.009 for both). However, interstitial edema was significantly reduced in the IR-BPC group compared to the IR group (*p* = 0.032) (Table 5, Figure 2).

Alveolar congestion also exhibited significant differences between the groups (*p* = 0.019). The IR group demonstrated a higher degree of alveolar congestion compared to the control and BPC groups (*p* = 0.004 and *p* = 0.012, respectively). In the IR-BPC group, alveolar congestion was significantly lower than in the IR group (*p* = 0.040) (Table 5, Figure 2).

Leukocyte infiltration (neutrophil/lymphocyte) was significantly different among the groups (*p* = 0.008). Leukocyte infiltration was more frequent in the IR group compared to the control and BPC groups (*p* = 0.002 and *p* = 0.006, respectively) (Table 5, Figure 2).

The thickness of the alveolar walls also differed significantly between the groups (*p* = 0.008). The IR group had significantly thicker alveolar walls compared to the control and BPC groups (*p* = 0.002 and *p* = 0.006, respectively) (Table 5, Figure 2).

In terms of the total damage score, significant differences were observed among the groups (*p* = 0.004). The total damage score was higher in the IR group compared to the control and BPC groups (*p* = 0.001 and *p* = 0.002, respectively). However, the total damage score was significantly reduced in the IR-BPC group compared to the IR group (*p* = 0.042) (Table 5, Figure 2).

No significant difference was found between the groups regarding interstitial hemorrhage (*p* = 0.292) (Table 5, Figure 2).

#### 3.2.3. Liver Tissue

Significant differences were observed among the groups in terms of hepatocyte degeneration (*p* = 0.030). Hepatocyte degeneration was more prominent in the IR group compared to the control and BPC groups (*p* = 0.010 for both) (Table 6, Figure 3).

Sinusoidal dilatation also differed significantly between the groups (*p* = 0.040). The IR group showed a higher degree of sinusoidal dilatation compared to the control and BPC groups (*p* = 0.013 for both). In contrast, sinusoidal dilatation in the IR-BPC group was significantly reduced compared to the IR group (*p* = 0.042) (Table 6, Figure 3).

Pyknosis (shrunken nuclei) was found to be significantly different among the groups (*p* = 0.025). Pyknosis was more frequently observed in the IR group compared to the control and BPC groups (*p* = 0.008 for both) (Table 6, Figure 3).

The number of necrotic cells was significantly different between the groups (*p* = 0.010). The IR group exhibited more necrotic cells compared to the control and BPC groups (*p* = 0.004 for both). However, the IR-BPC group showed a significantly lower number of necrotic cells compared to the IR group (*p* = 0.012) (Table 6, Figure 3).

Mononuclear (MN) cell infiltration in the parenchyma also showed significant differences among the groups (*p* = 0.027). The IR group had significantly higher levels of MN cell infiltration compared to the control and BPC groups (*p* = 0.009 for both). In the IR-BPC group, MN cell infiltration was significantly lower than in the IR group (*p* = 0.032) (Table 6, Figure 3).

## 4. Discussion

Ischemia typically occurs as a localized event but following the restoration of blood flow (reperfusion), various mediators from the affected tissue are released into the systemic circulation, triggering widespread effects on distant organs. Studies have shown that the systemic impact of ischemia–reperfusion (IR) injury arises from the activation of neutrophils, the complement cascade, and the release of proinflammatory and vasoactive mediators, including eicosanoids, NO, cytokines, and reactive oxygen species (ROS) [54,55,56]. These inflammatory and oxidative agents contribute to widespread endothelial dysfunction and immune activation.

The remote effects of IR injury are most frequently seen in the lungs, kidneys, liver, and cardiovascular systems, with the lungs often being the first organ to exhibit signs of damage due to their extensive vascular network [1]. This systemic involvement can lead to the development of a systemic inflammatory response syndrome (SIRS), which, when left unchecked, may progress to multiple organ dysfunction syndrome (MODS). It has been reported that up to 30–40% of mortality in intensive care units (ICUs) is linked to the systemic inflammatory consequences of IR injury, particularly in cases involving multi-organ failure [57]. The lungs, in particular, are highly susceptible to damage due to their role in filtering inflammatory mediators and activated neutrophils, which can exacerbate pulmonary edema, oxidative stress, and acute respiratory distress syndrome (ARDS) [10,58].

Moreover, recent studies have highlighted the role of mitochondrial dysfunction and endothelial permeability in promoting remote organ damage during IR injury, leading to further complications such as acute kidney injury (AKI) and hepatic necrosis [58]. Interventions aimed at modulating these systemic responses—through antioxidant therapies, inhibition of neutrophil infiltration, and targeted anti-inflammatory strategies—are critical to minimizing the mortality associated with IR-induced complications [54,59].

BPC 157, recognized for its potent antioxidant properties, contains four carboxylic groups, which are believed to play a crucial role in scavenging and neutralizing reactive oxygen species (ROS) [60]. Research has demonstrated that BPC 157 not only mitigates cellular damage induced by non-steroidal anti-inflammatory drugs (NSAIDs) [61,62] but also facilitates wound healing [63]. Additionally, it exhibits strong anti-inflammatory and antioxidant effects [36,37], providing protection to a variety of tissues [22,29,38,44,64,65,66,67,68], further supporting its therapeutic potential in conditions of oxidative stress and inflammation. It also has interference with the NO system activities and counteract harmful events arising from NO blockade [32]. It has a normalizing effect on NO tissue concentration and reduces both myeloperoxidase activity and oxidative stress [22].

In this study, we explored the effects of BPC 157 on three distinct tissues following an experimental ischemia–reperfusion (I/R) injury model, which involved clamping and subsequent declamping of the infra-renal aorta. The kidneys, liver, and lungs were selected for this study due to their heightened susceptibility to systemic inflammation and oxidative stress following lower extremity ischemia–reperfusion (IR) injury [4,8,9,10]. These organs possess extensive vascular networks and exhibit high metabolic activity, rendering them particularly vulnerable to IR-induced damage. As such, they serve as reliable indicators of systemic organ injury. Conversely, the heart and brain were not included in this study because of their distinct ischemia–reperfusion dynamics and intrinsic protective mechanisms. The brain, for instance, is exceptionally sensitive to ischemic insults due to its substantial metabolic demands and limited energy reserves, as it relies exclusively on glucose for adenosine triphosphate (ATP) production [69]. The heart, on the other hand, activates specialized cytoprotective signaling pathways, such as ischemic preconditioning and postconditioning, which mitigate reperfusion injury through mechanisms that differ significantly from those impacting other organs [69]. These organ-specific protective adaptations necessitate distinct experimental models and comprehensive investigative approaches. Specifically, studies involving the heart and brain require the inclusion of advanced molecular and genetic analyses to evaluate complex signaling pathways and accurately assess ischemia–reperfusion outcomes. Therefore, focusing on the kidneys, liver, and lungs allowed for a more targeted and systematic evaluation of BPC 157’s protective effects against systemic IR injury.

To the best of our knowledge, this is the first investigation that examines the impact of BPC 157 across multiple tissue types within a single experimental framework. This comprehensive approach provides a novel understanding of the therapeutic potential of BPC 157 in mitigating I/R injury in various tissues.

### 4.1. Effects of BPC 157 on Renal Tissue After Limb I/R

Limb I/R injury can result in significant damage to distant organs, with the kidneys being particularly susceptible. The systemic inflammatory response triggered by limb I/R releases inflammatory mediators and reactive oxygen species into the bloodstream, which in turn affects renal tissue [70]. Studies show that the accumulation of neutrophils and oxidative stress in the kidneys following limb I/R injury exacerbates renal dysfunction. The infiltration of proinflammatory cytokines, such as TNF-α and IL-6, leads to increased vascular permeability and cellular damage in renal tissues [71]. Additionally, oxidative stress caused by reactive oxygen species contributes to renal tissue injury by damaging cellular structures, including lipids, proteins, and DNA. This stress can lead to acute kidney injury following limb I/R injury, further complicating patient outcomes [72]. Protective interventions such as antioxidant therapies and ischemic preconditioning have been shown to reduce renal damage by modulating the inflammatory response and improving tissue oxygenation [73].

BPC 157, a stable gastric pentadecapeptide, has shown promising antioxidant and protective effects on renal tissue following ischemia–reperfusion (I/R) injury, a condition where restored blood flow after ischemia causes oxidative stress and tissue damage. BPC 157 exerts its effects primarily by reducing oxidative stress and inflammation, key contributors to renal injury in I/R scenarios. Studies have indicated that BPC 157 enhances the expression of antioxidant enzymes such as heme oxygenase-1 (HO-1) and heat shock proteins (HSPs 70 and 90), both of which play crucial roles in mitigating oxidative damage in renal tissue [74]. In experimental models of I/R injury, BPC 157 has been shown to reduce lipid peroxidation, a marker of oxidative stress, and protect renal cells from apoptosis. This is achieved through its ability to scavenge reactive oxygen species (ROS) and improve microcirculatory dynamics, thereby enhancing tissue oxygenation and preventing further oxidative damage [40]. In our study, the effects of BPC 157 on renal tissue following ischemia–reperfusion (I/R) injury in a limb were significant, and results reflected the possible antioxidative protective mechanism on renal tissue following ischemia–reperfusion of limbs. In terms of biochemical outcomes, the study revealed that the TAS levels in renal tissue were markedly lower in the IR group compared to both the control and BPC 157-treated groups. Interestingly, the TAS levels in the IR-BPC group were significantly higher than those in the IR group (*p* = 0.019), suggesting that BPC 157 enhanced antioxidant defenses. Conversely, TOS was significantly elevated in the IR group but was notably lower in the IR-BPC group (*p* = 0.014), indicating a reduction in oxidative stress due to BPC 157. Similarly, OSI, a marker of oxidative damage, was higher in the IR group but significantly decreased in the IR-BPC group (*p* < 0.001). Additionally, PON1 enzyme activity, which is crucial for antioxidant defense, was substantially reduced in the IR group but partially restored in the IR-BPC group (*p* < 0.001).

BPC 157 modulates the inflammatory response by inhibiting the production of pro-inflammatory cytokines like TNF-α and IL-6, which are elevated during I/R injury [65,66,75]. The ability of BPC 157 to stabilize blood vessels and promote angiogenesis has also been observed [17,63], further supporting its protective role in renal I/R injury. This vasculoprotective effect ensures better perfusion and oxygen delivery to the affected renal tissue, reducing the extent of ischemic damage and facilitating faster recovery of renal function [76]. These protective effects of BPC 157 were also present histopathologically in our study. The IR group showed significant glomerular vacuolization, tubular dilation, vascular vacuolization and hypertrophy, and Bowman space dilation, all of which were considerably reduced in the IR-BPC group. For instance, tubular dilation in the IR group was more prominent than in the control and BPC groups (*p* = 0.023), but BPC 157 significantly ameliorated this damage (*p* = 0.046). Additionally, tubular cell shedding and glomerular vacuolization were significantly lower in the IR-BPC group compared to the untreated IR group (*p* = 0.012 and *p* = 0.027, respectively). These results underscore BPC 157’s role in reducing both biochemical oxidative stress markers and histopathological damage, highlighting its potential as a therapeutic agent for mitigating renal injury following limb ischemia–reperfusion. The histological findings of our study demonstrated that renal tissue with BPC 157 was subjected to less anti-inflammatory response after IR of limb, compared to IR group.

Our study, alongside previous research on renal tissue, collectively highlights that BPC 157 exerts profound renoprotective effects in cases of remote organ injury following limb ischemia–reperfusion. It significantly reduces oxidative stress, mitigates tissue damage, and enhances antioxidant defenses within renal tissue.

### 4.2. Effects of BPC 157 on Lung Tissue After Limb I/R

Limb I/R injury has a profound impact on the lungs. The reperfusion phase triggers the release of inflammatory mediators and reactive oxygen species (ROS), which play a key role in driving lung tissue damage. Lung injury following limb I/R is characterized by acute inflammation, oxidative stress, and increased vascular permeability, which can lead to pulmonary edema, neutrophil infiltration, and impaired gas exchange. Studies have reported that inflammatory cytokines such as TNF-α and IL-1β play critical roles in exacerbating lung tissue damage after limb I/R injury [77]. Histopathological changes include thickening of the alveolar walls, interstitial edema, and the presence of hyaline membranes, which are indicative of acute lung injury [78]. Moreover, the oxidative stress generated during reperfusion leads to increased malondialdehyde (MDA) levels in lung tissues, reflecting the extent of lipid peroxidation and tissue damage. This condition is further aggravated by the recruitment of neutrophils, which release proteolytic enzymes and ROS, contributing to additional tissue injury. Various studies have shown that treatments aimed at reducing oxidative stress, such as antioxidants like N-acetylcysteine and melatonin, can mitigate lung damage by scavenging ROS and reducing inflammation [79,80]. Additionally, preconditioning strategies such as remote ischemic preconditioning (RIPC) have been demonstrated to alleviate lung injury by modulating inflammatory pathways and enhancing antioxidant defenses [81].

BPC 157 has demonstrated notable antioxidant and cytoprotective effects on lung tissue, particularly in the context of ischemia–reperfusion (I/R) injury. As a distant organ, the lungs are often affected by limb I/R injury, which leads to systemic inflammation and oxidative stress. BPC 157 is thought to counter these harmful effects through several mechanisms. It has been shown to reduce oxidative stress by scavenging reactive oxygen species (ROS) and boosting the activity of antioxidant enzymes, which help in protecting lung tissues from lipid peroxidation and oxidative damage [44]. By modulating inflammatory pathways, BPC 157 suppresses the excessive release of pro-inflammatory cytokines such as TNF-α and IL-1β [40,82,83]. Through this mechanism, it has the potential to prevent further damage to the lung parenchyma. Our study supports the data found in the literature, reflecting the antioxidant effects of BPC 157 on lung tissue after IR of limbs. In terms of antioxidant and oxidative stress markers, the TAS in lung tissue was significantly reduced in the IR group compared to the control and BPC 157-treated groups in our study. The TAS levels in the BPC 157-treated group were notably higher than in the untreated IR group (*p* = 0.017), indicating that BPC 157 bolstered the lung’s antioxidant defenses. Conversely, TOS levels were significantly elevated in the IR group but were lower in the BPC 157-treated group (*p* < 0.001). This reduction in TOS points to a decrease in oxidative stress, suggesting that BPC 157 plays a protective role in minimizing oxidative damage. The OSI, a measure of the balance between oxidants and antioxidants, was also significantly lower in the BPC 157 group compared to the IR group (*p* < 0.001). Furthermore, PON1 activity, an enzyme crucial for antioxidant defense, was significantly reduced in the IR group but partially restored in the BPC 157 group (*p* = 0.013).

Regarding the peptide’s role in promoting angiogenesis and stabilizing endothelial function, BPC 157 has been observed to support microcirculation and enhance the oxygen supply to tissues [40]. This function helps mitigate the damage caused by ischemia–reperfusion injury, including in lung tissue, by improving endothelial stability and reducing oxidative stress [40,64]. In our study, histological analysis revealed that the IR group displayed severe lung tissue damage, including increased interstitial edema, alveolar congestion, leukocyte infiltration, and thickening of the alveolar walls. However, BPC 157 significantly attenuated these pathological changes. For instance, the severity of interstitial edema was significantly reduced in the BPC 157 group compared to the IR group (*p* = 0.027). Alveolar congestion was also less pronounced in the BPC 157-treated group compared to the untreated IR group (*p* = 0.019). The infiltration of leukocytes, which reflects inflammation, was significantly reduced in the BPC 157-treated group (*p* = 0.008), indicating that BPC 157 has strong anti-inflammatory effects. The total damage score, which incorporates these histopathological parameters, was significantly lower in the BPC 157 group (*p* = 0.004), demonstrating its overall protective effect on lung tissue. To conclude, in the group treated with BPC 157, we observed macroscopically that they were less affected by oxidative stress and the inflammatory response, highlighting the potential protective effect of BPC 157.

### 4.3. Effects of BPC 157 on Liver Tissue After Limb I/R

Liver tissue, as a distant organ, is highly susceptible to I/R injury following limb ischemia. When reperfusion is restored, a systemic inflammatory response is triggered, resulting in oxidative stress and damage to the liver. Several studies have shown that limb I/R injury can lead to significant hepatocellular injury marked by elevated levels of proinflammatory cytokines and oxidative stress markers such as malondialdehyde (MDA) [84]. The liver undergoes oxidative stress due to an influx of reactive oxygen species (ROS), which cause lipid peroxidation, protein oxidation, and DNA damage. Additionally, inflammatory mediators, including TNF-α and IL-6, are upregulated, contributing to endothelial dysfunction and hepatocellular apoptosis [70]. Histopathologically, liver tissue exposed to limb I/R injury exhibits congestion, cellular necrosis, and infiltration of inflammatory cells. These changes are also associated with elevated markers of oxidative stress and impaired antioxidant defense, as seen in decreased glutathione (GSH) levels and glutathione peroxidase (GPx) activity [85]. Furthermore, ischemic preconditioning has shown promise in reducing liver damage by enhancing the liver’s antioxidant capacity and reducing inflammatory responses [86]. These protective strategies highlight the importance of early intervention in mitigating liver injury as a distant effect of limb ischemia–reperfusion injury.

BPC 157, a stable gastric pentadecapeptide, has demonstrated potential antioxidant and protective effects on liver tissue following ischemia–reperfusion (I/R) injury. The liver, as a distant organ, is often vulnerable to damage during limb I/R injury due to the systemic release of inflammatory mediators and reactive oxygen species (ROS). BPC 157’s antioxidant properties have been shown to mitigate this oxidative stress by scavenging ROS and reducing lipid peroxidation, which are crucial mechanisms in preventing hepatocellular injury [83]. Biochemically, BPC 157 increases the expression of antioxidant enzymes such as heme oxygenase-1 (HO-1) and NO synthase (NOS-3), which play key roles in enhancing the liver’s defense against oxidative damage. By reducing the levels of pro-inflammatory cytokines such as TNF-α and IL-6, BPC 157 helps in attenuating inflammation and promoting tissue repair [74]. BPC 157 significantly mitigated oxidative stress markers in liver tissue in our study. TAS was substantially lower in the IR group compared to the control and BPC 157-treated groups. The administration of BPC 157 significantly elevated TAS levels in the IR group (*p* = 0.002), indicating enhanced antioxidant activity. Conversely, TOS, which was elevated in the IR group, was markedly reduced with BPC 157 treatment (*p* < 0.001), pointing to a reduction in oxidative damage. Additionally, OSI was significantly lower in the BPC 157-treated group compared to the untreated IR group (*p* < 0.001). Similarly, PON1 enzyme activity, which is critical for mitigating oxidative stress, was significantly higher in the BPC 157-treated group compared to the IR group (*p* < 0.001).

Histopathological studies have reported that BPC 157 reduces congestion, cellular necrosis, and sinusoidal dilatation in liver tissue exposed to I/R injury, thereby preventing extensive liver damage [44]. Histological analysis of our study revealed that the IR group exhibited considerable liver damage, characterized by hepatocyte degeneration, sinusoidal dilatation, and necrosis. BPC 157 treatment significantly reduced the severity of these histopathological changes. For instance, hepatocyte degeneration in the BPC 157-treated group was significantly lower compared to the I/R group (*p* = 0.030). Additionally, sinusoidal dilatation was markedly decreased in the BPC 157-treated group compared to the I/R group (*p* = 0.040). Necrotic cells, a clear indicator of tissue damage, were significantly reduced in the BPC 157-treated group (*p* = 0.010). Mononuclear cell infiltration, which signifies an inflammatory response, was also significantly lower in the BPC 157-treated group (*p* = 0.027). Taken together with data from the literature, our results show that BPC 157 has a potential antioxidant and anti-inflammatory effect on liver damage after limb ischemia–reperfusion.

While this study provides comprehensive insights into the protective effects of BPC 157 on distant organ damage following ischemia–reperfusion (IR) injury, several limitations should be acknowledged. First, this study was conducted on an animal model (rats), which, while useful for understanding fundamental physiological and pathological processes, may not fully capture the complexity of human IR injury. Therefore, the direct clinical applicability of these findings requires further validation through translational and clinical studies. Second, the experimental design included a relatively short reperfusion period (120 min), which may not reflect the full progression of IR injury or the long-term protective effects of BPC 157. Future studies with extended reperfusion periods are necessary to evaluate the sustained impact of BPC 157 on tissue regeneration and functional recovery. Third, the study exclusively examined the kidneys, liver, and lungs—organs known to be highly susceptible to systemic IR injury—without investigating other critical organs such as the brain and heart. Future studies should broaden the scope to include these organs to provide a more comprehensive understanding of BPC 157’s systemic protective effects. Fourth, although the study assessed oxidative stress and histopathological changes, it did not include molecular markers of inflammation, such as proinflammatory cytokines (e.g., TNF-α, IL-6, IL-1β). Incorporating cytokine profiling via ELISA or multiplex assays would offer deeper insights into BPC 157’s anti-inflammatory mechanisms and should be considered in future research. Fifth, the study focused on a single dosage of BPC 157 based on prior studies. However, evaluating multiple dosage groups would help identify the optimal therapeutic window, ensuring both efficacy and safety. Dose–response studies are warranted to optimize treatment strategies and minimize the risk of under- or over-treatment. Lastly, while key oxidative stress markers (TAS, TOS, OSI, and PON-1) were measured, the analysis could be expanded by including additional antioxidant enzymes such as glutathione peroxidase (GPx), superoxide dismutase (SOD), and catalase. Moreover, integrating molecular and genetic analyses, including the assessment of signaling pathways (e.g., NF-κB, iNOS), would provide a more detailed understanding of the cellular mechanisms underlying BPC 157’s protective effects. As our research provide foundational evidence of BPC-157’s protective effects, addressing these limitations in future research with dose escalation studies will offer a more comprehensive evaluation of BPC 157’s therapeutic potential in ischemia–reperfusion injury.

## 5. Conclusions

In conclusion, this study has demonstrated that BPC 157, a stable gastric pentadecapeptide, offers significant protective effects against IR injury across multiple distant organs, including the kidneys, lungs, and liver. Biochemical and histopathological analyses confirmed that BPC 157 mitigates oxidative stress, reduces inflammation, and enhances antioxidant defenses in tissues subjected to I/R injury. These findings highlight the therapeutic potential of BPC 157 in reducing damage from systemic inflammation and oxidative stress, which are key contributors to multi-organ dysfunction following I/R injury. While the results are promising, further research, particularly in clinical settings, is needed to validate the efficacy of BPC 157 in human models and to explore its long-term protective effects. Expanding future studies to include a broader range of organ systems and employing molecular and genetic analyses will provide a more comprehensive understanding of BPC 157’s mechanisms of action, paving the way for its potential clinical applications in treating I/R-related injuries.

## Figures and Tables

**Figure 1 medicina-61-00291-f001:**
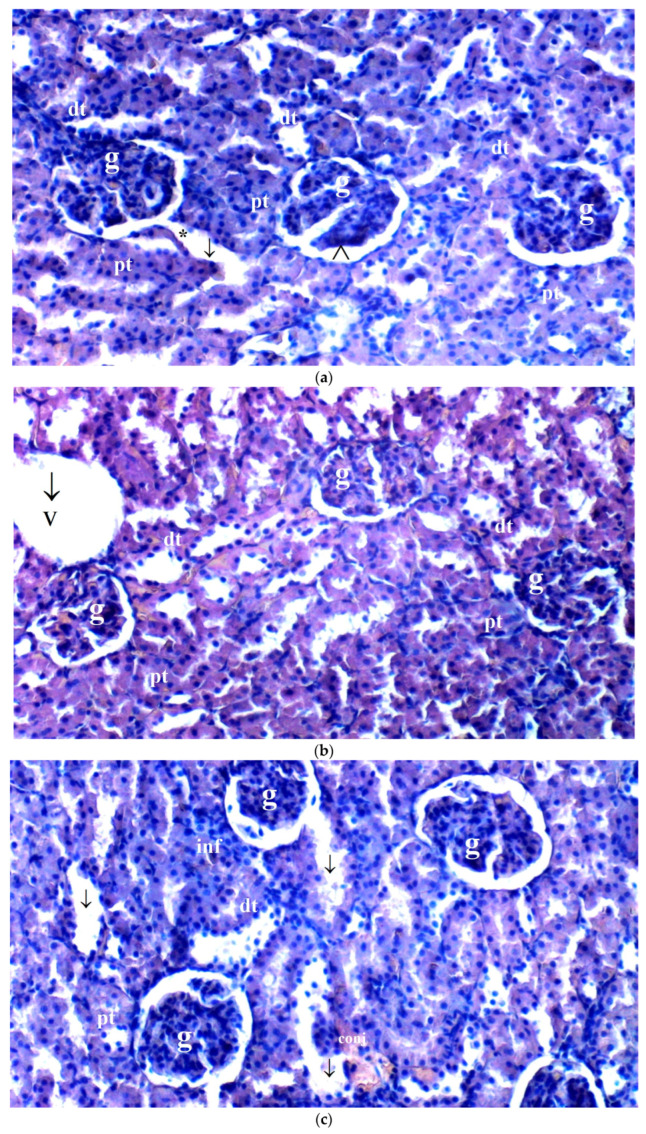
(**a**–**d**) Histopathological examination of renal tissues of C, B, IR and IR-BPC groups. (**a**): Normal renal tissue in group C, (**b**): renal tissue of group B, (**c**): renal tissue of group IR with severe degeneration and changes, (**d**): renal tissue of group IR-BPC with mild degeneration. Abbreviations: dt—distal tubule, pt—proximal tubule, g—glomerulus, *— degenerated glomerulus, ∧— Bowman’s space, ↓—dilated tubules, v— vacuolation, inf—inflammation, conj—congestion.

**Figure 2 medicina-61-00291-f002:**
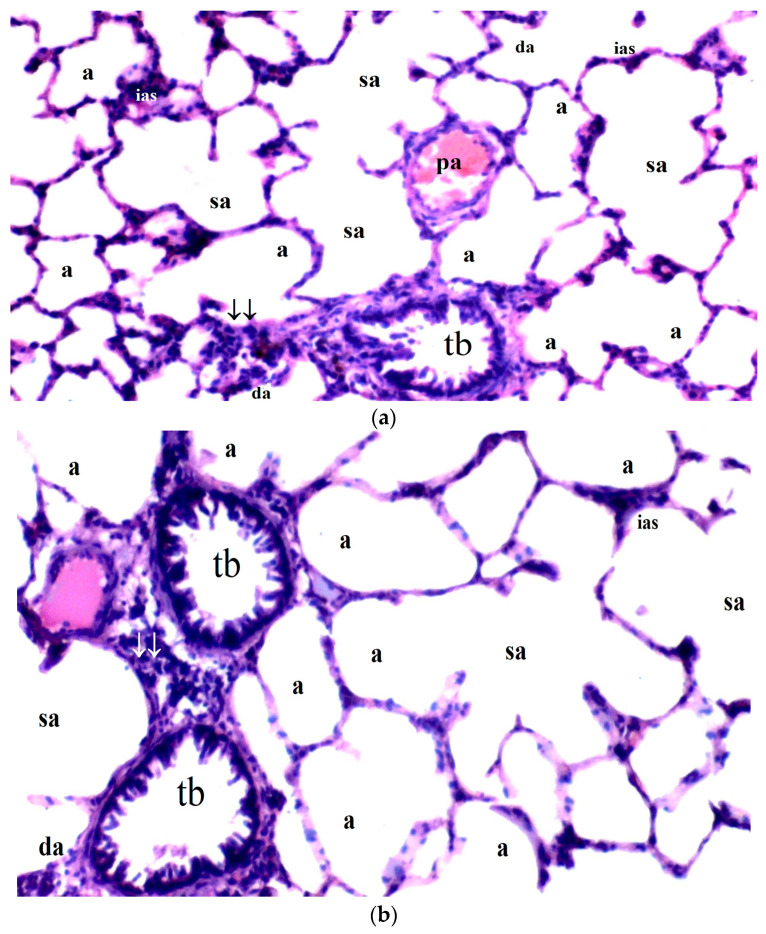
(**a**–**d**) Histopathological examination of lung tissues of C, B, IR and IR-BPC groups. (**a**): Normal lung tissue in group C, (**b**): lung tissue of group B, (**c**): lung tissue of group IR with increased leukocyte infiltration and thicker of the alveolar walls, (**d**): lung tissue of group IR-BPC with mild degeneration compared to group IR. Abbreviations: a—alveolus, sa—saccus alveolaris, da—ductus alveolaris, conj—congestion, tb—terminal bronchiole, ias—interalveolar septum, pa—pulmonary artery, ↓↓—parenchyma, rb—respiratory bronchiole, cong—capillary congestion.

**Figure 3 medicina-61-00291-f003:**
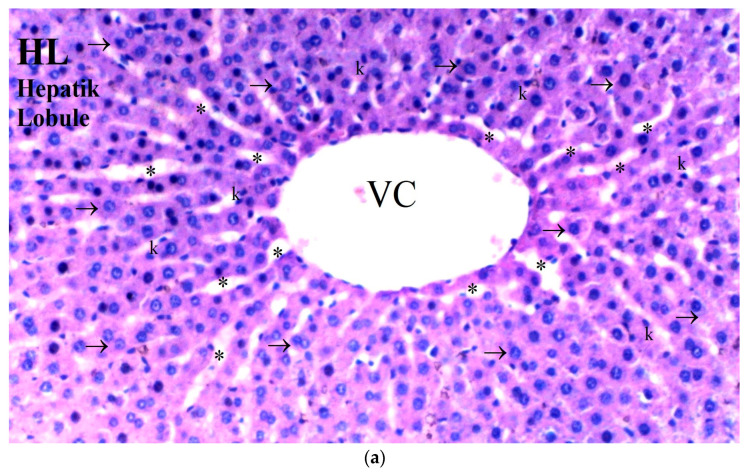
(**a**–**d**) Histopathological examination of liver tissues of C, B, IR and IR-BPC groups. (**a**): Normal liver tissue in group C, (**b**): liver tissue of group B, (**c**): liver tissue of group IR with increased necrotic cells and hepatocyte degeneration, (**d**): liver tissue of group IR-BPC with mild degeneration compared to group IR. Abbreviations: ep—epithelium, vc—vena centralis, hl—hepatic lobule, →—hepatocytes, k—Kupffer cells, *—sinusoids, (*)—sinusoid dilatation, cong—capillary congestion, ↓—parenchyma.

**Table 1 medicina-61-00291-t001:** Biochemical analyses of the renal tissue (mean ± SE).

	Group C (n = 6)	Group B (n = 6)	Group IR (n = 6)	Group IR-BPC (n = 6)	*p* Value (ANOVA)
Total antioxidant status (mmol/L)	1.48 ± 0.08	1.41 ± 0.03	1.26 ± 0.03 *^,^**	1.39 ± 0.03 ***	0.019
Total oxidative status (μmol/L)	13.71 ± 1.13	15.47 ± 1.39	20.54 ± 1.41 *^,^**	16.17 ± 0.48 ***	0.014
Oxidative Stress Index	0.93 ± 0.07	1.10 ± 0.09	1.57 ± 0.13 *^,^**	1.23 ± 0.05 *^,^***	<0.001
Paraoxonase 1 (U/L)	11.30 ± 0.90	9.01 ± 0.26	5.67 ± 0.50 *^,^**	7.62 ± 0.70 *^,^***	<0.001

* *p* < 0.05: Compared to group C, ** *p* < 0.05: compared to group B, *** *p* < 0.05: compared to group IR.

**Table 2 medicina-61-00291-t002:** Biochemical analyses of the lung tissue (mean ± SE).

	Group C (n = 6)	Group B (n = 6)	Group IR (n = 6)	Group IR-BPC (n = 6)	*p* Value (ANOVA)
Total antioxidant status (mmol/L)	1.69 ± 0.08	1.57 ± 0.06	1.37 ± 0.05 *^,^**	1.58 ± 0.06 ***	0.017
Total oxidative status (μmol/L)	23.50 ± 0.24	23.56 ± 1.89	32.65 ± 0.49 *^,^**	26.76 ± 1.32 ***	<0.001
Oxidative Stress Index	1.41 ± 0.07	1.53 ± 0.08	2.39 ± 0.06 *^,^**	1.81 ± 0.11 *^,^**^,^***	<0.001
Paraoxonase 1 (U/L)	7.88 ± 0.83	7.22 ± 0.35	4.64 ± 0.54 *^,^**	7.10 ± 0.81 ***	0.013

* *p* < 0.05: Compared to group C, ** *p* < 0.05: compared to group B, *** *p* < 0.05: compared to group IR.

**Table 3 medicina-61-00291-t003:** Biochemical analyses of the liver tissue (mean ± SE).

	Group C (n = 6)	Group B (n = 6)	Group IR (n = 6)	Group IR-BPC (n = 6)	*p* Value (ANOVA)
Total antioxidant status (mmol/L)	2.09 ± 0.03	2.06 ± 0.04	1.83 ± 0.06 *^,^**	1.92 ± 0.06 *^,^**	0.002
Total oxidative status (μmol/L)	21.07 ± 0.60	23.26 ± 1.00	27.94 ± 0.87 *^,^**	22.79 ± 1.01 ***	<0.001
Oxidative Stress Index	1.01 ± 0.03	1.13 ± 0.06	1.54 ± 0.06 *^,^**	1.19 ± 0.05 *^,^***	<0.001
Paraoxonase 1 (U/L)	44.71 ± 2.35	43.79 ± 3.28	28.21 ± 1.27 *^,^**	36.30 ± 1.28 *^,^**^,^***	<0.001

* *p* < 0.05: Compared to group C, ** *p* < 0.05: compared to group B, *** *p* < 0.05: compared to group IR.

**Table 4 medicina-61-00291-t004:** Histopathological analysis scores of the renal tissue (mean ± SE).

	Group C (n = 6)	Group B (n = 6)	Group IR (n = 6)	Group IR-BPC (n = 6)	*p* Value (ANOVA)
Glomerular vacuolization	0.50 ± 0.22	0.50 ± 0.22	1.50 ± 0.22 *^,^**	0.67 ± 0.21 ***	0.012
Tubular dilatation	0.50 ± 0.22	0.50 ± 0.22	1.67 ± 0.33 *^,^**	0.83 ± 0.31 ***	0.023
Vascular vacuolization and hypertrophy	0.50 ± 0.22	0.50 ± 0.22	1.33 ± 0.21 *^,^**	0.67 ± 0.21 ***	0.040
Tubular cell degeneration and necrosis	0.50 ± 0.22	0.67 ± 0.21	1.33 ± 0.21	0.67 ± 0.21	0.121
Bowman space dilatation	0.33 ± 0.21	0.33 ± 0.21	1.67 ± 0.33 *^,^**	0.50 ± 0.22 ***	0.003
Tubular hyaline casts	0.33 ± 0.21	0.50 ± 0.22	1.33 ± 0.21 *^,^**	0.67 ± 0.21 ***	0.019
Lymphocyte infiltration	0.33 ± 0.21	0.33 ± 0.21	1.17 ± 0.17	0.67 ± 0.21	0.071
Tubular cell shedding	0.33 ± 0.21	0.33 ± 0.21	1.17 ± 0.17 *^,^**	0.50 ± 0.22 ***	0.027

* *p* < 0.05: Compared to group C, ** *p* < 0.05: compared to group B, *** *p* < 0.05: compared to group IR.

**Table 5 medicina-61-00291-t005:** Histopathological analysis scores of the lung tissue [mean ± SE].

	Group C (n = 6)	Group B (n = 6)	Group IR (n = 6)	Group IR-BPC (n = 6)	*p* Value (ANOVA)
Interstitial edema	0.33 ± 0.21	0.33 ± 0.21	1.17 ± 0.17 *^,^**	0.50 ± 0.22 ***	0.027
Interstitial hemorrhage	0.33 ± 0.21	0.33 ± 0.21	0.83 ± 0.17	0.50 ± 0.22	0.292
Alveolar congestion	0.33 ± 0.21	0.50 ± 0.22	1.33 ± 0.21 *^,^**	0.67 ± 0.21 ***	0.019
Leukocyte infiltration (neutrophil/lymphocyte)	0.17 ± 0.17	0.33 ± 0.21	1.33 ± 0.21 *^,^**	0.83 ± 0.31	0.008
Thickness of the alveolar walls	0.17 ± 0.17	0.33 ± 0.21	1.33 ± 0.21 *^,^**	0.83 ± 0.31	0.008
Total damage score	1.33 ± 0.88	1.67 ± 1.05	6.00 ± 0.52 *^,^**	3.33 ± 0.92 ***	0.004

* *p* < 0.05: Compared to group C, ** *p* < 0.05: compared to group B, *** *p* < 0.05: compared to group IR.

**Table 6 medicina-61-00291-t006:** Histopathological analysis scores of the liver tissue [mean ± SE].

	Group C (n = 6)	Group B (n = 6)	Group IR (n = 6)	Group IR-BPC (n = 6)	*p* Value (ANOVA)
Hepatocyte degeneration	0.33 ± 0.21	0.33 ± 0.21	1.33 ± 0.33 *^,^**	0.67 ± 0.21	0.030
Sinusoidal dilatation	0.50 ± 0.22	0.50 ± 0.22	1.33 ± 0.21 *^,^**	0.67 ± 0.21 ***	0.040
Pyknosis	0.33 ± 0.21	0.33 ± 0.21	1.17 ± 0.17 *^,^**	0.67 ± 0.21	0.025
The number of necrotic cells	0.33 ± 0.21	0.33 ± 0.21	1.33 ± 0.21 *^,^**	0.50 ± 0.22 ***	0.010
Mononuclear cell infiltration in the parenchyma	0.33 ± 0.21	0.33 ± 0.21	1.17 ± 0.17 *^,^**	0.50 ± 0.22 ***	0.027

* *p* < 0.05: Compared to group C, ** *p* < 0.05: compared to group B, *** *p* < 0.05: compared to group IR.

## Data Availability

Data from the current study are available from the corresponding author upon request.

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
