# Peer review of "Protective Effects of BPC 157 on Liver, Kidney, and Lung Distant Organ Damage in Rats with Experimental Lower-Extremity Ischemia–Reperfusion Injury"

_medicina, 2025, doi:10.3390/medicina61020291_

Round 1

Reviewer 1 Report

Comments and Suggestions for Authors

The paper is focused on the effects of BPC 157 on organ damage at the ischemia-reperfusion. The limited understanding of the intricate mechanisms underlying ischemia-reperfusion injury hinders the development of effective therapeutic interventions.

This is an interesting paper; however there are several crucial points that need consideration before the story can be accepted for publication:

- Introduction is difficult to follow. There are many organ damages at the ischemia-reperfusion. The Authors should describe why did they investigate only these organs (kidneys, lungs, liver), and not the brain or heart, for example.

In the chapter Material and Methods there is no detailed description of the histological assessment of tissues.

The article contains two figures 1, the quality of the figures should be improved.

To draw conclusions, it is not enough to assess oxidative stress and histology alone. Why didn't the authors perform an ELISA test for proinflammatory cytokines.

Author Response

Reviewer 1

The paper is focused on the effects of BPC 157 on organ damage at the ischemia-reperfusion. The limited understanding of the intricate mechanisms underlying ischemia-reperfusion injury hinders the development of effective therapeutic interventions. 

This is an interesting paper; however, there are several crucial points that need consideration before the story can be accepted for publication:

1-) Introduction is difficult to follow. There are many organ damages at the ischemia-reperfusion. The Authors should describe why did they investigate only these organs (kidneys, lungs, liver), and not the brain or heart, for example.

2-) In the chapter Material and Methods there is no detailed description of the histological assessment of tissues.

3-) The article contains two figures 1, the quality of the figures should be improved.

4-) To draw conclusions, it is not enough to assess oxidative stress and histology alone. Why didn't the authors perform an ELISA test for proinflammatory cytokines.

We sincerely appreciate the opportunity to revise and resubmit our manuscript titled "Protective Effects of BPC 157 on Liver, Kidney, and Lung Distant Organ Damage in Rats with Experimental Lower Extremity Ischemia-Reperfusion Injury."

We are deeply grateful for the insightful and constructive feedback provided by the reviewers. Their comments have been instrumental in refining and strengthening our manuscript. Below, we present detailed responses to each reviewer’s comment, along with a summary of the corresponding revisions, which are highlighted in yellow in the revised manuscript.

1-)

We have revised the Introduction to enhance clarity and coherence. Specifically, we have provided a more comprehensive context for ischemia-reperfusion (IR) injury and elaborated on the role of BPC 157 in mitigating distant organ damage. We have emphasized the rationale for focusing on the kidneys, liver, and lungs. Additionally, new references have been incorporated into the third paragraph, cited as references (10) and (11).

The selection of the kidneys, liver, and lungs was based on their heightened vulnerability to systemic inflammation and oxidative stress induced by lower extremity ischemia-reperfusion injury. These organs are particularly susceptible due to their extensive vascularization and high metabolic activity, making them reliable indicators of systemic IR damage. In contrast, the heart and brain were not included in this study because their distinct protective mechanisms and differing ischemia-reperfusion dynamics necessitate specialized experimental designs and comprehensive investigative approaches. The rationale for prioritizing the lungs, kidneys, and liver has been further elaborated in the Discussion section.

“The kidneys, liver, and lungs were selected for this study due to their heightened susceptibility to systemic inflammation and oxidative stress following lower extremity ischemia-reperfusion (IR) injury (4,6–8). These organs possess extensive vascular networks and exhibit high metabolic activity, rendering them particularly vulnerable to IR-induced damage. As such, they serve as reliable indicators of systemic organ injury. Conversely, the heart and brain were not included in this study because of their distinct ischemia-reperfusion dynamics and intrinsic protective mechanisms. The brain, for instance, is exceptionally sensitive to ischemic insults due to its substantial metabolic demands and limited energy reserves, as it relies exclusively on glucose for adenosine triphosphate (ATP) production (60). The heart, on the other hand, activates specialized cytoprotective signaling pathways, such as ischemic preconditioning and postconditioning, which mitigate reperfusion injury through mechanisms that differ significantly from those impacting other organs (60). These organ-specific protective adaptations necessitate distinct experimental models and comprehensive investigative approaches. Specifically, studies involving the heart and brain require the inclusion of advanced molecular and genetic analyses to evaluate complex signaling pathways and accurately assess ischemia-reperfusion outcomes. Therefore, focusing on the kidneys, liver, and lungs allowed for a more targeted and systematic evaluation of BPC 157's protective effects against systemic IR injury.”

2-)

Revised

3-)

Revised

4-)

We acknowledge that assessing proinflammatory cytokines (e.g., TNF-α, IL-6, IL-1β) through ELISA would have provided valuable insights into the inflammatory response associated with ischemia-reperfusion injury. However, due to resource constraints, ELISA assays were not incorporated into this study. This limitation has been explicitly addressed in the revised Limitations section, and we have proposed its inclusion in future studies to further elucidate the anti-inflammatory mechanisms of BPC 157.

Reviewer 2 Report

Comments and Suggestions for Authors

This study offers a detailed exploration of the protective effects of BPC-157 against distant organ damage in a rat model of lower extremity ischemia-reperfusion (I/R) injury. The comprehensive documentation of both histopathological and biochemical findings provides valuable insight into the compound's protective mechanisms across multiple organ systems, including the liver, kidneys, and lungs. However, a few aspects warrant further consideration:

1. The study focuses on oxidative stress markers and histopathological changes. Incorporating additional molecular markers of inflammation, such as cytokine profiles (e.g., TNF-α, IL-6), could offer a more nuanced understanding of BPC-157's anti-inflammatory properties and potential toxicities.

2. The proposed antioxidant and anti-inflammatory mechanisms of BPC-157 could be further substantiated. Would additional biopsies and molecular analyses help validate these pathways and clarify the compound's role in treatment efficacy and resistance mechanisms?

3. While the histopathological assessments were detailed, the biochemical analysis could be expanded. Including a broader range of oxidative stress indicators and enzymatic assays, such as glutathione peroxidase (GPx) or catalase, might provide a clearer picture of the oxidative balance.

4. The antioxidant effects of BPC-157 are highlighted, yet the study does not explore the influence of varying dosages. Would assessing multiple dosage groups help define the optimal therapeutic range and reduce the risk of under- or over-treatment?

Author Response

Reviewer 2

This study offers a detailed exploration of the protective effects of BPC-157 against distant organ damage in a rat model of lower extremity ischemia-reperfusion (I/R) injury. The comprehensive documentation of both histopathological and biochemical findings provides valuable insight into the compound's protective mechanisms across multiple organ systems, including the liver, kidneys, and lungs. However, a few aspects warrant further consideration:

  1. The study focuses on oxidative stress markers and histopathological changes. Incorporating additional molecular markers of inflammation, such as cytokine profiles (e.g., TNF-α, IL-6), could offer a more nuanced understanding of BPC-157's anti-inflammatory properties and potential toxicities.
  2. The proposed antioxidant and anti-inflammatory mechanisms of BPC-157 could be further substantiated. Would additional biopsies and molecular analyses help validate these pathways and clarify the compound's role in treatment efficacy and resistance mechanisms?
  3. While the histopathological assessments were detailed, the biochemical analysis could be expanded. Including a broader range of oxidative stress indicators and enzymatic assays, such as glutathione peroxidase (GPx) or catalase, might provide a clearer picture of the oxidative balance.
  4. The antioxidant effects of BPC-157 are highlighted, yet the study does not explore the influence of varying dosages. Would assessing multiple dosage groups help define the optimal therapeutic range and reduce the risk of under- or over-treatment?

We sincerely appreciate the opportunity to revise and resubmit our manuscript titled "Protective Effects of BPC 157 on Liver, Kidney, and Lung Distant Organ Damage in Rats with Experimental Lower Extremity Ischemia-Reperfusion Injury."

We are deeply grateful for the insightful and constructive feedback provided by the reviewers. Their comments have been instrumental in refining and strengthening our manuscript. Below, we present detailed responses to each reviewer’s comment, along with a summary of the corresponding revisions, which are highlighted in yellow in the revised manuscript.

1-)

We agree that measuring proinflammatory cytokines would provide more comprehensive insights into the anti-inflammatory effects of BPC-157. Although this study primarily focused on oxidative stress parameters and histological evaluations, we acknowledge the significance of assessing cytokines such as TNF-α and IL-6 through ELISA to better characterize the inflammatory response. Due to resource limitations, cytokine profiling was not conducted; however, this limitation has been addressed in the revised Discussion section. We plan to incorporate cytokine assays in future studies to enable a more in-depth analysis of BPC-157's immunomodulatory mechanisms.

2-)

We appreciate this valuable suggestion and acknowledge that molecular analyses, such as immunohistochemistry for nuclear factor kappa B (NF-κB), inducible nitric oxide synthase (iNOS), and antioxidant enzymes, could further strengthen our findings. However, the primary objective of this study was to provide foundational evidence of BPC-157’s protective effects. This recommendation has been incorporated into the Limitations section, and we propose that future research should include molecular pathway analyses to more comprehensively elucidate the mechanisms underlying BPC-157’s therapeutic actions.

3-)

We agree that including a broader range of oxidative stress biomarkers would strengthen the study. Although we assessed total antioxidant status (TAS), total oxidant status (TOS), oxidative stress index (OSI), and paraoxonase-1 (PON-1), resource limitations constrained the scope of biomarker evaluation. As this study aimed to provide foundational evidence of BPC-157’s protective effects, we have acknowledged this limitation in the revised Limitations section and emphasized the need for future studies to incorporate a wider array of oxidative stress markers.

4-)

We acknowledge the importance of investigating the dose-response relationship of BPC-157. In this study, we selected a dosage of 20 μg/kg based on previous literature to maintain consistency and comparability with existing research. However, we agree that evaluating multiple dosage levels could help determine the optimal therapeutic range and minimize the risks of under- or over-treatment. This limitation has been addressed in the revised manuscript, and we propose future dose-escalation studies to establish the most effective and safe dosage of BPC-157.

Round 2

Reviewer 1 Report

Comments and Suggestions for Authors

Thank you for the corrections. However, in the conclusions the authors report on the anti-inflammatory effect of the drug. At the same time, pro-inflammatory cytokines were not studied, the conclusion was made only on the basis of histological and oxidative studies.